# communications
# engineering

# Neural network based prediction of the efficacy of ball milling to separate cable waste materials

Jiaqi Lu [1,2✉], Mengqi Han[2], Shogo Kumagai[1✉], Guanghui Li[2] & Toshiaki Yoshioka[1]

Material recycling technologies are essential for achieving a circular economy while reducing greenhouse gas emissions. However, most of them remain in laboratory development. Machine learning (ML) can promote industrial application while maximising yield and environmental performance. Herein, an asynchronous-parallel recurrent neural network was developed to predict the dynamic behaviour when separating copper and poly(vinyl chloride) components from the cable waste. The model was trained with six datasets (treatment conditions) at 3600 epochs. High accuracy was confirmed based on a mean-square error of 0.0015–0.0145 between the prediction and experimental results. The quantitative relationship between the input features and the separation yield was identified using sensitivity analysis. The charged weight of cables and impact energy were determined as the critical factors affecting the separation efficiency. The ML framework can be widely applied to recycling technologies to reveal the process mechanism and establish a quantitative relationship between process variables and treatment outputs.

[1] Graduate School of Environmental Studies, Tohoku University, 6-6-07 Aoba, Aramaki-aza, Aoba-ku, Sendai, Miyagi 980-8579, Japan. [2] Innovation Centre for Environment and Resources, Shanghai University of Engineering Science, No.333 Longteng Road, Songjiang District, Shanghai 201620, China. ✉email: wilsherelu@foxmail.com; kumagai@tohoku.ac.jp

The life cycle of a material is commonly associated with high resource depletion, greenhouse gas (GHG) emissions, and other environmental impacts. Achieving sustainable development goals requires promoting the advance of material recycling technologies as an indispensable component of the circular economy. Many laboratory (lab)-scale recycling technologies have been proposed for various types of wasted materials, e.g., rare-earth elements[1], polymers[2], and resources in electronic[3] and photovoltaic panel[4] waste. However, the industrial application of lab-scale recycling technologies is confronted with the high cost of scaling-up tests and the uncertainty of environmental impacts from the process and consumed resources[5]. Thus, modelling recycling technologies with machine learning (ML) can be a promising tool to predict potential consumption and recovered materials on an industrial scale based on experimental data[6,7], revealing the optimal process design and environmental benefits.

This study focused on an emerging mechanical approach for separating copper (Cu) and poly(vinyl chloride) (PVC) from electric cable waste[8–10]. The generation of global e-waste will reach 74.7 million tonnes (Mt) by 2030[11], with a considerable portion being wires and cables[12]. Improper plastic and metal waste management may cause environmental pollution, such as ecotoxicity[13] and microplastic issues[14]. Widely applied energy recovery process (e.g., incineration with power generation) also complicate the mitigation of GHG emissions[15]. Thus, developing an advanced recycling technology is essential to simultaneously recover pure materials (i.e., Cu and PVC) with high efficiency and accuracy.

There are many lab-scale processes for material recovery from cable at lab-scale[16], such as rotational moulding[17], flotation[18], and freezing[19]. Our proposed approach consists of de-plasticising the PVC covering and a ball milling process for the high-accuracy separation (summarised in Supplementary Notes 1 of Supplementary Information (SI)), where the separation mechanism (Supplementary Notes 2 in SI) was revealed using the discrete element method (DEM)[20]. Compared with the widely commercialised nugget process with low Cu purity[21], the separation accuracy can reach over 99% of Cu purity. Although the organic solvent was used in the process, only plasticisers will dissolve in the solvents (which can be recycled by distillation[10]), and the PVC resin remains in a solid state. Unlike that PVC is dissolved by the organic solvent into a mixed wax state (vinyloop process)[22], the solid-state PVC is more satisfying for removing the organic solvent and recycling. The feasibility was also validated based on experiments with a bench-scale mill[23].

This process has potential applications in the recycling of PVC cable waste from household electrical appliances, such as televisions and computers[24]. With appropriate pre-treatment, this could also be applied to cable mixed in construction[25] and automobile[26] waste. In terms of high-voltage transmission cables, this process could be modified with a different solvent because the covers are commonly made of polyethylene[27]. To advance such industrial applications, a simulation approach can demonstrate the separation mechanism[28]; however, predicting the separation under various conditions and treatment times is currently not possible because of the lack of a dynamic model. Based on the literature review summarised in Table S1 of SI, many efficient processes for cable separation have been developed at a lab scale; however, process modelling was rare, especially for yield prediction by ML.

ML is a data-driven tool that automatically and efficiently determines parameters for a complex mathematic model[29]. Apart from its wide application in artificial intelligence[30], ML can support the development of recycling technologies by predicting a product with various input features[31–33]. However, in the current application for modelling the recycling technologies, ML was used as a big data model based on the complex algorithms without considering the process mechanism. Consequently, establishing a robust ML model for recycling technologies often requires enormous data[34]. Furthermore, the treatment time is considered an input feature without a dynamic mechanism[6]. Thus, an advanced model for including the time series during the recycling process and considering the process mechanism should be developed with limited lab-scale data to accelerate the application of technology. Existing studies modelled cable separation processes as polynomial[35] or support vector regression[36]; however, none of these works carry out dynamic modelling of the process, which is innovatively realised by recurrent neural network (RNN) in this study. Previously, we have developed an RNN model combined with the chemical reaction mechanism for predicting the dechlorination degree of PVC in ethylene glycol/NaOH solvent[37]. However, the separation of PVC and Cu from cable waste is a physical process that requires a more general modelling framework for predicting the dynamic behaviour without chemical kinetics.

In this study, a modified RNN was developed to model and predict the dynamics of the separation. ML was not directly used to predict the final separation yield based on the input features (experimental conditions) and treatment duration, but to predict the separation rate under various ball milling conditions during a short timeframe. Because the basic RNN method carries out parameter updating based on the real-time prediction loss, to avoid overfitting based on the end stage of treatment, different calculation timesteps were introduced in our model. Thus, training and predictions were performed simultaneously for all datasets (different experimental conditions and results) with different timesteps, which is known as parallel and asynchronous learning.

To accomplish this work, the input features, including the cable properties and milling conditions of grinding balls, were refined from our previous lab-scale experiments[20]. Then, the influence of input features on separation rate was quantified with defined multi-layer RNNs. With a linear piecewise function based on the time-dependent separation yields from the experiments as training data, the parameters in the networks were successfully optimised by simultaneous training for all datasets with various timesteps (i.e., asynchronous-parallel). The overall framework of the asynchronous-parallel RNN model is depicted in Fig. 1a–f, where the detailed explanation can be found in the section of model framework in methods. The "black box" of the trained model was examined with a sensitivity analysis to identify the quantitative relationship between the separation yields and input features, which can be explained by the process mechanism. The proposed ML algorithm can be widely applied to a lab-scale process for predicting the output and optimising the design, revealing the potential cost and environmental performance for industrial applications.

## Results

**Visualised training process to build a robust model**. Model training is the primary process in ML for determining the optimal combination of parameters with the smallest error between prediction and given experimental data. The change in $Y_{sep}$ over the treatment time by the average and fluctuation range (calculated by standard deviation) based on ten training iterations is illustrated in Fig. 2a–f to represent the internal operations of the training process. Although $Y_{sep}$ during the training process is predicted, it is not the final prediction result because the model parameters are dynamically updated in real time. When the

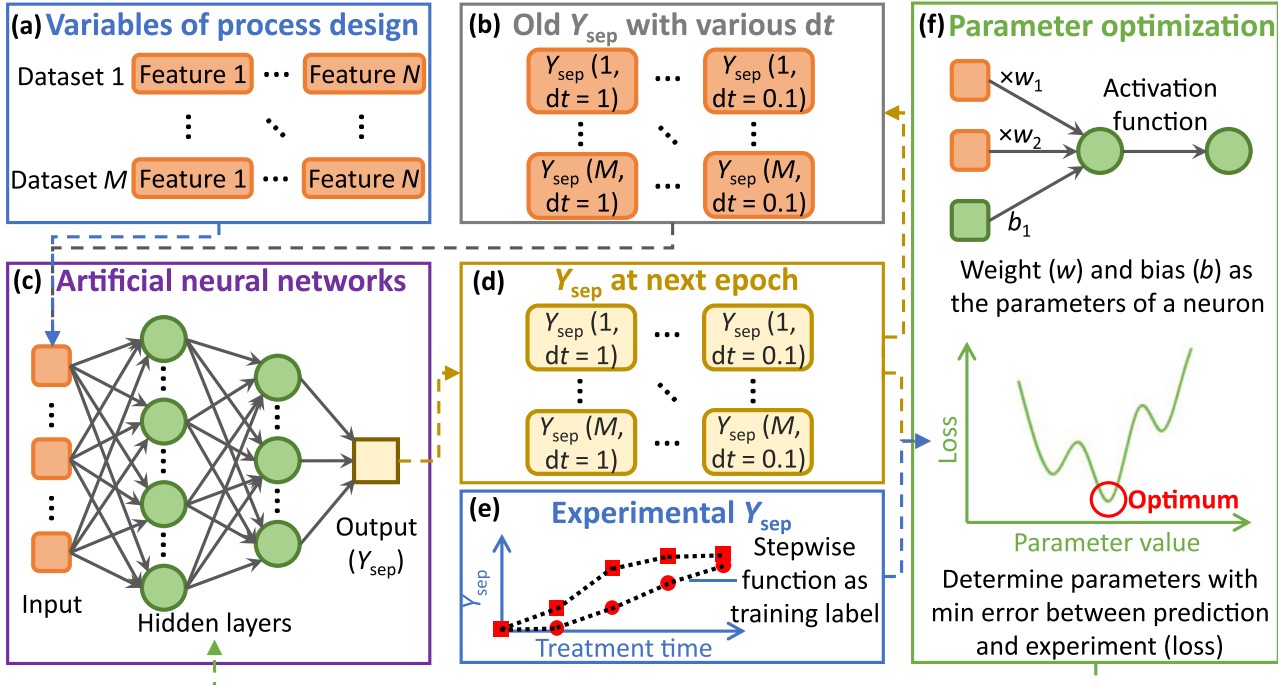

**Fig. 1 Framework of the proposed asynchronous-parallel RNN model explaining varibale input and output during modeling training.** Input variables (orange round rectangles): **a** the datasets with different process designs; **b** the old separation yield ($Y_{sep}$) at last epoch. The RNN model (**c**) for calculating the $Y_{sep}$ at next epoch (**d**). With a stepwise function based on experimental $Y_{sep}$ as training labels (**e**), the parameters (green circles) in the RNN model can be optimised (**f**).

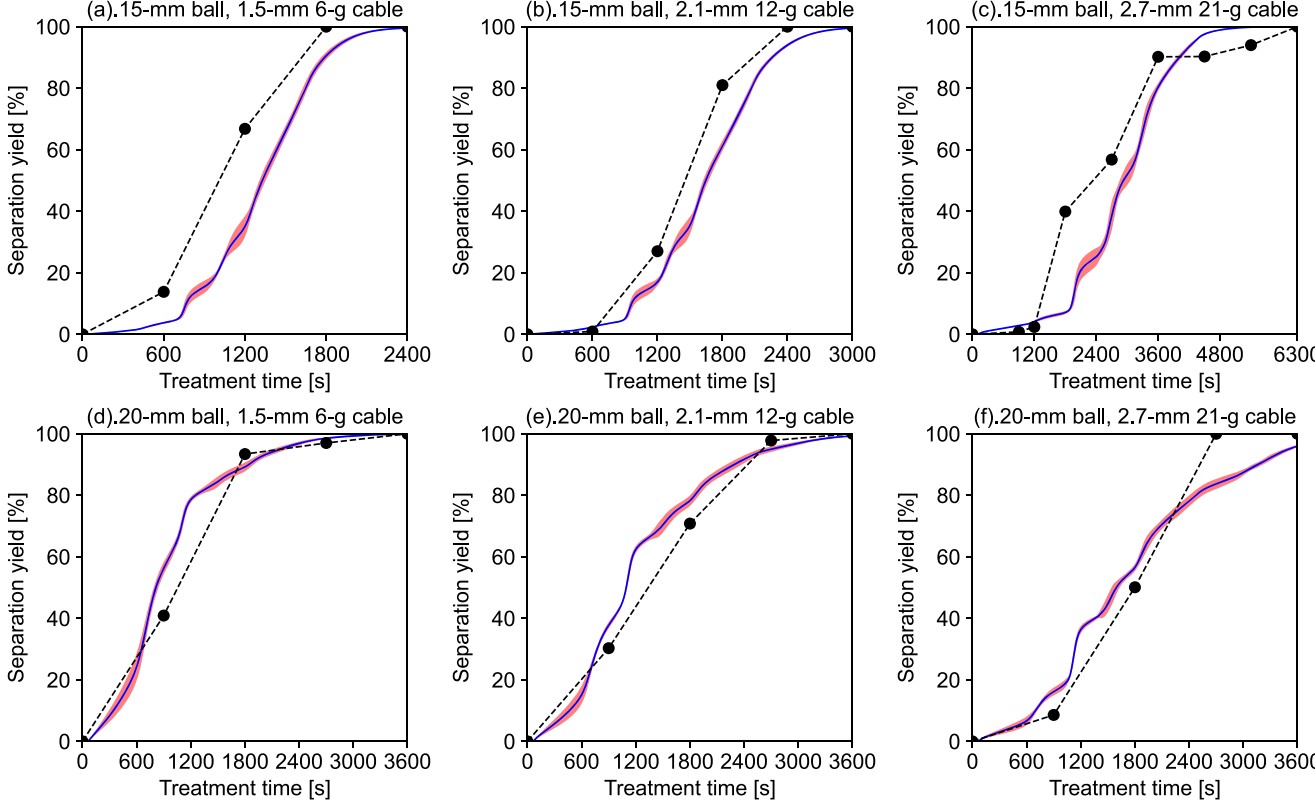

**Fig. 2 Visualisation of the training process to optimise model parameters by plotting the dynamic changing of separation yield ($Y_{sep}$) versus treatment time.** **a**–**c** varying cable diameter and weight with fixed 15-mm ball (entries 1–3); **d**–**f** varying cable diameter and weight with fixed 20-mm ball (entries 4–6). The black-circle point represents the experimental separation yield ($Y_{sep-exp}$) linked by a dashed line representing training labels, while the blue line represents the average values of $Y_{sep}$ from 10-time training. The red area represents the fluctuation range based on the standard deviation of $Y_{sep}$. For clarity, subplots (**a**–**f**) are illustrated separately for each dataset even though they are simultaneously trained in the model.

model is successfully trained and the parameters are fixed, the prediction function can be executed.

The unsmooth increase in $Y_{sep}$ during the training process is illustrated in Fig. 2. Based on the separation mechanism, some time is required to form a crack throughout the cable covering after a ball-to-cable collision. Thus, at the beginning of treatment, the increase in $Y_{sep}$ is minimal, especially for Entry 3 with the 15-mm ball and 2.7-mm cable. When $Y_{sep}$ starts to increase, the parameters cannot be immediately optimised to reflect the rapid growth, leading to an incremental error. However, the parameters can be gradually updated by a gradient as a function of input features toward the elevated $r_{sep}$. Furthermore, if $r_{sep}$ is overestimated, the parameters will be updated in the opposite direction. Using the stepwise functions formed by the experimental data as the training label will also cause a dramatic change during some periods because of the considerable noise compared with the actual values. To avoid the influence of such noise and experimental errors on the training, a dropout method which randomly omits neurons in hidden layers can be adopted to mitigate overfitting[38].

Because each experimental sample is simultaneously trained with asynchronous timesteps (Fig. 2 only illustrates the training at the highest d$t$), the parameters should be balanced to represent all treatment conditions and the entire time series. The red areas in each subplot of Fig. 2 are based on the standard deviation of $Y_{sep}$ for 10-time training, where a larger area reveals periods with higher uncertainty. From a global perspective, the uncertainty of the training process is low because the red areas are limited. The relative uncertainty is concentrated in the mid-term for the treatment with 15-mm balls, while 20-mm balls will result in more training fluctuations in the early and late stages. To reduce this uncertainty, a better sampling-interval design can be suggested. For example, increasing the data points around the starting point of the separation when both the $Y_{sep}$ and $r_{sep}$ change sharply. Meanwhile, experiments with a scaled-up ball mill also reduce the randomness of ball-to-cable collision. Because the model can be successfully trained to derive robust parameters, the next step is to validate the prediction with fixed model parameters.

**Validation of prediction accuracy with trained model**. With the successfully trained model, the time series prediction of $Y_{sep}$ throughout the treatment time was conducted for all conditions, as depicted in Fig. 3a–f. Unlike during the training process, the model parameters are fixed during the validation (and application); thus, there is no fluctuation in the predicted $Y_{sep}$ after multiple runs. The mean-square error (MSE) and coefficient of determination ($R^2$) are also calculated based on the predicted and experimental $Y_{sep}$ values (rather than the stepwise functions) to validate the model. A lower MSE and higher $R^2$ (close to 1) represent more accurate predicted results consistent with the experimental data.

The prediction error is low because all MSEs are <0.015, while the dynamic behaviour of $Y_{sep}$ under various treatment conditions is well-fitted by the model (most of $R^2 > 0.95$). Compared with the training process, the changing curves of $Y_{sep}$ over the treatment time are smoother with the fixed model parameters. The prediction errors are relatively lower for separation of the 1.5-mm and 2.1-mm cables, while considerable gaps exist between the prediction and experiment with the 2.7-mm cables. Because the experimental data has error and uncertainty caused by the randomness of ball-to-cable collision, it is challenging to reflect this phenomenon with the current model. For example, the growth of $Y_{sep\text{-}exp}$ almost stopped from $t = 3600$ s to $t = 4500$ s, re-increasing to 1 until $t = 6300$ s. In contrast, the trained model is not overfitted into a tortuous increase curve of $Y_{sep}$ to match

the noise from experimental fluctuation. With the prediction performance validated for the established model, the next step is to analyse how the input features affect the dynamic change in $Y_{sep}$.

**Insight of trained model based on sensitivity analysis**. The quantitative relationship between five input features (along with the modelled relationship between ball diameter and impact energy) and $Y_{sep}$ are examined with a sensitivity analysis to avoid the transparency issue of trained models with a large number of parameters. Figure 4 illustrates the comparison of $Y_{sep}$ under various input features during a rapid growth period ($t = 1800$ s). Concerning the influence of cable properties (Fig. 4a–c), the $Y_{sep}$ is more sensitive to the charged weight than cable density and diameter. The more-charged cables caused a greater allocation of collision intensity, resulting in a decline in $Y_{sep}$. This phenomenon can also be supported by the discrete element method simulation on the different charging ball mill ratio[39]. Despite the minor influence, the increase in cable diameter negatively affects the separation. In principle, a lower density will cause a decreased diameter for a certain weight and length of cables, and it has been reported that small particles have less contact chance compared with large ones[40]. For thicker cables, more collision power is necessary to crush the PVC covering. The density of a cable should vary according to the PVC and Cu composition, which could result in different dynamic separation behaviours. However, the existing experimental findings and sensitivity analysis regarding the impact of cable density are insufficient to conclude a clear pattern.

The influence of mechanical factors on the ball milling process is investigated in Fig. 4d–f. While maintaining constant impact energy, the increase in ball diameter will lead to a slight decrease in $Y_{sep}$. The increase in ball diameter is not equivalent to the enhanced mechanical efficiency of ball milling[41]. Based on the spatial structure of a sphere array, a larger diameter will expand the gap volume among the grinding balls—consequently, the collision probability and $r_{sep}$ decrease. As depicted in Fig. 4e, a non-linear change in $Y_{sep}$ occurs when varying the impact energy. Sufficient impact energy is mandatory to crush the PVC covering through an inelastic collision. Nevertheless, excess impact energy disperses the cable motion according to momentum conservation and increases the amount of Cu wire crushed into unrecyclable powder.

Thus, an optimal ball milling design exists for the efficient and high-quality recovery of the PVC and Cu. It is hard to change only the ball diameter with fixed impact energy because impact energy calculations are based on the kinetic energy of grinding balls decided by weight and velocity[42]. Thus, the relationship between ball diameter and the impact energy was fitted by a cubic function (Fig. S3 in SI) to conduct a more realistic sensitivity analysis (Fig. 4f). The change in $Y_{sep}$ in Fig. 4f is a comprehensive effect based on the single change in ball diameter and impact energy in Fig. 4d, e. The smaller gap space of small balls can offset the insufficient impact energy to enhance $Y_{sep}$. The optimal values of ball diameter are changed for 2.1-mm and 2.7-mm cables compared with those in Fig. 4e. Therefore, the optimal ball milling process design is vital for separating PVC and Cu from cable waste with complex properties.

**Discussion**
This study proposes an advanced RNN framework for developing solid waste recycling technology. Solid waste commonly has high heterogeneity in substance composition, composite structure, particle size, and physical and chemical properties. Furthermore, there is considerable uncertainty in the design parameters and operating conditions for a lab-scale recycling process. Mathematical modelling assisted by ML is a promising strategy to solve

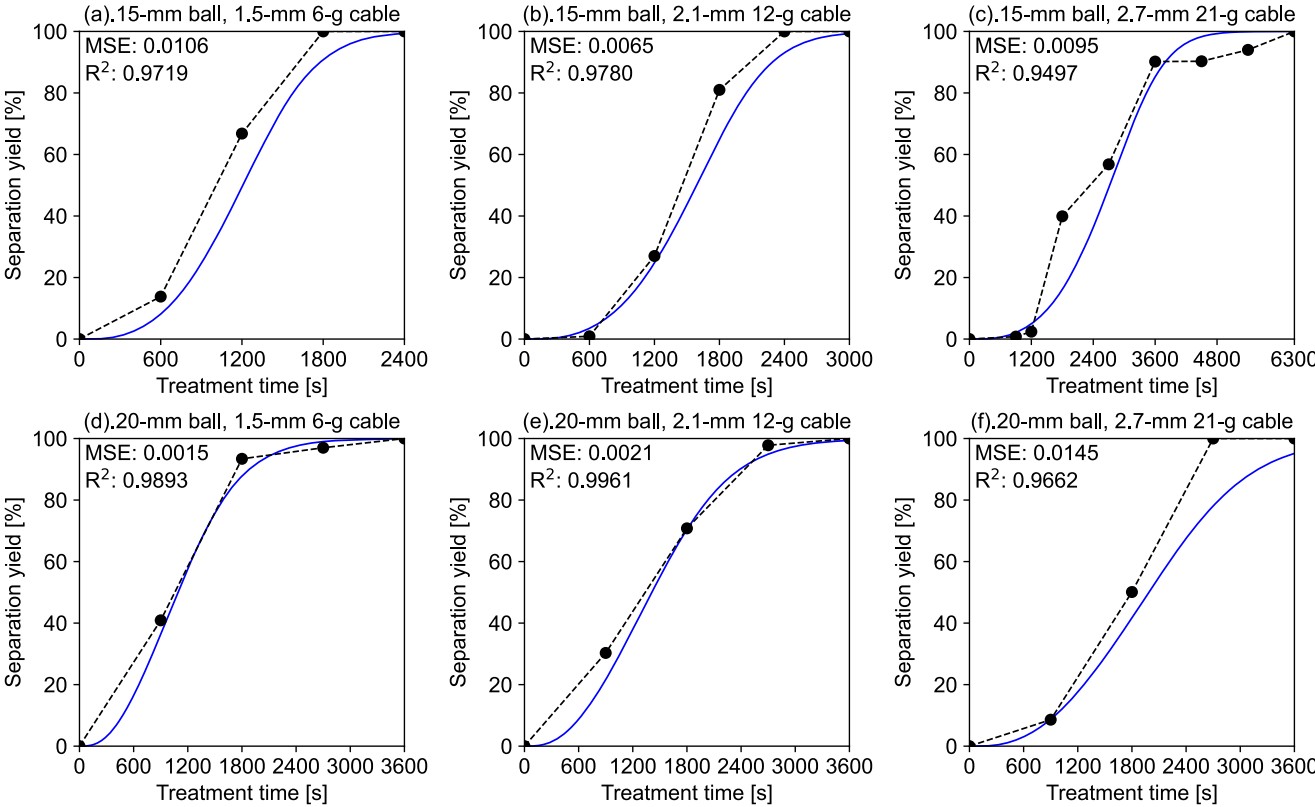

**Fig. 3 Comparison between $Y_{sep\text{-}exp}$ and the predicted $Y_{sep}$ versus treatment time for all experimental conditions based on the trained model with optimised parameters. a–c** varying cable diameter and weight with fixed 15-mm ball (entries 1–3); **d–f** varying cable diameter and weight with fixed 20-mm ball (entries 4–6). MSE and $R^2$ are calculated based on $Y_{sep\text{-}exp}$ and the predicted $Y_{sep}$ at the same treatment time (excluding the zero point).

this type of complex problem (with multi-dimensional variables) because it quantifies the influence of process variables (input features) on material recovery efficiency (predicted output). The non-linear model can be optimised to suggest an energy-saving, environmentally-friendly, economically-feasible design for emerging recycling technologies[43], which is the ultimate target of our research. However, for the current result shown in Fig. 4, only the local-optimal solution can be obtained for the maximum $Y_{sep}$ with a fixed treatment time and cable type. In the future, the applicability of the developed model for different process designs and ball milling scales should be validated.

Although prediction accuracy is not the priority because this is the first application of the proposed model to mechanical treatment for cable separation, we summarised some potential improvements to reduce prediction error. First, more experimental data during the accumulation period of impact energy are favourable for accurately predicting the start point of separation. Furthermore, the increase in throughput with an up-scale reactor can reduce errors from the experiment. The complexity of the model during construction can be enhanced by pretreating the input features (e.g., applying the power function), increasing the hidden layers, and introducing other activation functions (e.g., ReLU, tanh, or their combination[44]). Such techniques may increase the probability of a suitable non-linear relationship between input features and predicted results. However, the overfitting issue should be avoided when expanding the datasets and model complexity.

The new findings can be combined with the previous experimental results to derive critical points for the industrial application of PVC and Cu recovery from cable waste by ball milling. Because the charged cable weight and impact energy are critical factors for separation, the increase in impact energy (e.g., by increasing the rotation speed) should be accompanied when raising the throughput.

A moderate rotation speed may also exist to maximise the ball milling efficiency[45]. For various cable specifications, the optimal mechanical conditions differ in terms of separation efficiency. In general, the optimal conditions vary based on the sample properties[46,47]. Given that cable waste of various sizes is frequently mixed, a straightforward solution is to combine grinding balls with various diameters. Furthermore, intelligent operation can be achieved by sampling the cable properties of each waste batch and predicting the optimal conditions and required treatment time based on the developed ML model with appropriate modifications. Meanwhile, the ball milling conditions (e.g., rotation speed) can be dynamically changed to generate sufficient impact energy in the early and middle stages. The impact energy at the ending period can then be reduced to avoid the crushing of Cu.

**Prospect of industrial application.** The annual generation of e-waste was 53.6 Mt in 2019[11]; meanwhile, it was reported that ~10 wt.% is of cable waste[48]. Because the applicability for the complex and mixed cable waste was confirmed[10], the industrial application of the proposed high-accuracy cable separation process can facilitate the recycling of 5.4 Mt of cable waste. Assuming a 3:1 of PVC-to-Cu weight ratio based on the density, 4.1 and 1.3 Mt of PVC and Cu can be annually recycled, which is equivalent to 7.5% and 5.3% of the produced virgin materials[49,50], respectively. For sustainability, this technology contributes to the circular economy associated with plastic and metal and creates more value-added employment opportunities. Instead of the unsustainable but sanitised landfilling and incineration in developed countries, the manual stripping and open burning still exist in developing countries[51]. The promotion of cable waste recycling is indispensable for establishing a material circulation system associated with e-waste, bringing more high-income employment.

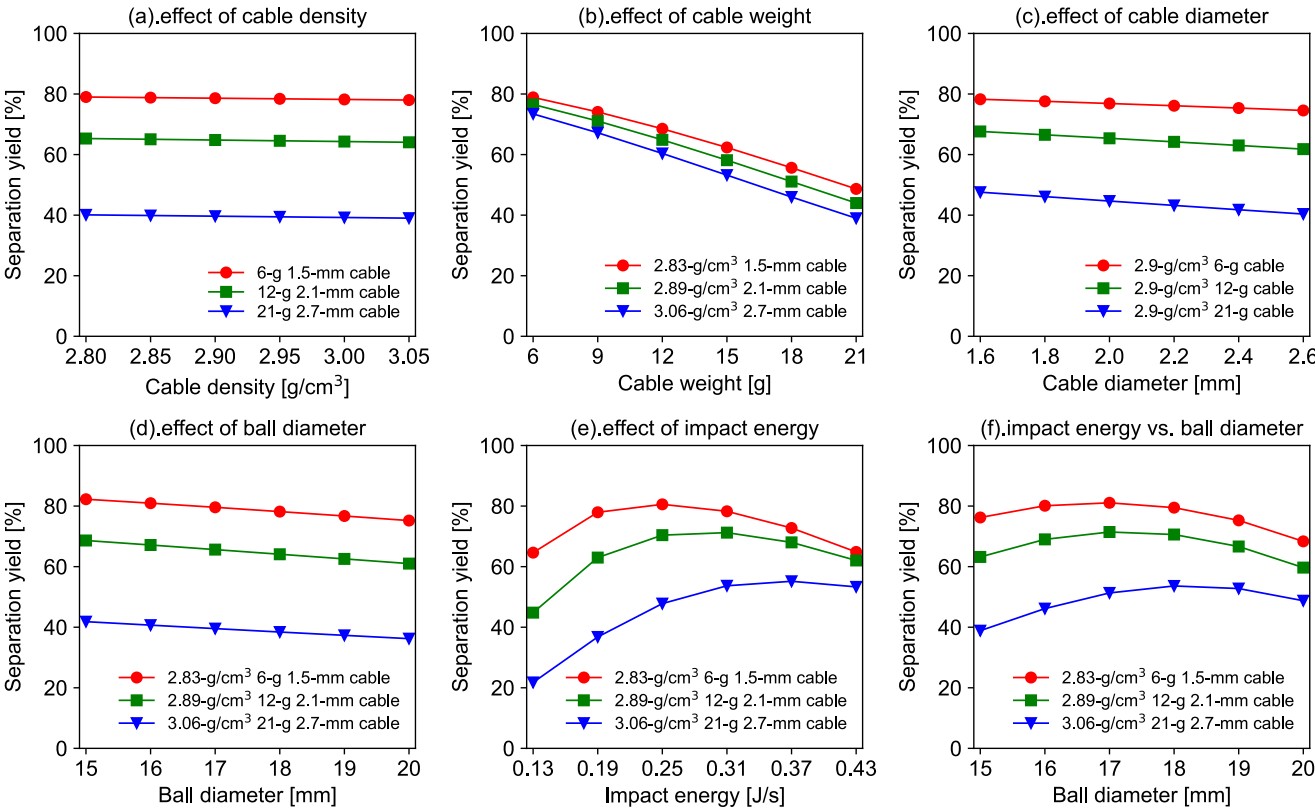

**Fig. 4 Sensitivity analysis revealing the effect of the input features on the $Y_{sep}$ at $t$ = 1800 s (30 min). a** cable density, **b** cable weight, **c** cable diameter, **d** ball diameter, **e** impact energy, and **f** modelled impact energy versus ball diameter based on Supplementary Notes 3 in SI. For the calculation of subplots (**a–f**), the ball diameter and impact energy are fixed at 17.5 mm and 0.2 J/s if neither are investigated as a variable.

In terms of the environmental impacts, the industrial application of cable waste recycling can avoid the ecosystem contaminant (heavy metals and persistent organic pollutants) and GHG emissions from thermal treatments[52,53]. However, the cable separation process causes additional energy consumption and potential pollution owing to the use of organic solvent. This study aims to develop a prediction model for the industrial application. The precise control of ball milling time and design optimisation will definitely reduce the electricity consumption. Meanwhile, by screening different solvents[54], butyl acetate is preferred not for the highest efficiency, but for the non-toxic and non-hazardous properties[55]. Herein, a one-pot process was successfully developed for reducing the exposure of organic solvent[9].

Considering GHG emissions, the recycling of 1-kg PVC or Cu will avoid 2.50 and 6.45 kg $CO_2$-e during the cradle-to-gate production of virgin materials based on the ecoinvent database[56]. For a bench-scale ball mill reactor, a 50-min treatment is necessary for achieving complete separation of 65-g cable waste[23], in which the throughput and energy efficiency are not optimised. Assuming the operating power of ball milling as 100 W[57], the electricity consumption is 1.28 kWh/kg separated cable waste, resulting in a 0.85 kg $CO_2$-e/kg separated cable waste of GHG emissions based on the current power grid of Japan. Thus, even though the lab-scale treatment efficiency is low, the avoided GHG emissions would be far higher than the additionally produced ones. Overall, up to 14.05 Mt GHG can be globally mitigated when the separation is widely applied. However, this calculation does not include the solvent supplement, which must be investigated in our future research.

The scaling-up potential is an essential issue for the industrial application. The ball milling process consumes considerable electricity, and the financial cost is high[58], which hinders the

pilot-scale operation test. First, the electricity consumption can also be simulated based on the DEM method[59], where the energy efficiency can be ex-ante assessed before the industrial application. However, the DEM simulation on a large-scale ball mill requires substantial computation time; so, the ML can also be used for predicting the electricity consumption. Furthermore, the process design and operation could be simultaneously optimised through a non-linear programming method for minimising financial cost[60]. To prevent the human exposure and cost increase, the organic solvent loss should be mitigated by controlling operation conditions (e.g., pressure and temperature) and installing a vapour recovery system[61].

In terms of a classic RNN model for time series prediction, typically only one series of datasets is trained at a time[62]. The simultaneous training with multiple series of datasets can facilitate the model to learn general parameters applicable to different series[63]. However, there is a phenomenon that the model is overfitted based on data from the later period[37]. Thus, the asynchronous training with multiple timesteps can mitigate the overfitting issue, which can be proven by validating the trained model with fewer timestep numbers shown in Figs. S5–7 in SI compared with Fig. 3. In addition, the validation of the trained model with one timestep in Fig. S5, which can be regarded as a conventional RNN, elucidates that the prediction cannot be accomplished without asynchronous training. The developed RNN model can be applied to other ball milling processes, even to general industrial processes affected by dynamic factors. For example, ball milling can facilitate the mechanochemical treatment of solid waste[64,65] which commonly operates as a batch reactor. The RNN model can be adapted to predict chemical reaction rates under various temperatures, reactant concentrations, and other parameters. Moreover, ML has been widely applied in computational fluid dynamics to improve simulation efficiency and accuracy[66,67].

**Table 1 Datasets of input features based on the previously reported experiments[20].**

| Entry | Ball diameter [mm] | Cable diameter [mm] | Cable weight [g] | Cable density [g/cm³] | Impact energy [J/s] | Data numbers for error calculation |
|-------|--------------------|--------------------|--------------------|--------------------|--------------------|--------------------|
| 1 | 15 | 1.5 | 6 | 2.83 | 0.147 | 4 |
| 2 | 15 | 2.1 | 12 | 2.89 | 0.156 | 5 |
| 3 | 15 | 2.7 | 21 | 3.06 | 0.129 | 8 |
| 4 | 20 | 1.5 | 6 | 2.83 | 0.376 | 4 |
| 5 | 20 | 2.1 | 12 | 2.89 | 0.421 | 4 |
| 6 | 20 | 2.7 | 21 | 3.06 | 0.468 | 4 |

The multi-timestep training method proposed in this study has the potential to be applied in such algorithms to improve parameter optimisation.

## Conclusions

An asynchronous-parallel RNN framework was developed to model the complex operation of waste recycling technologies with multi-dimensional variables. The algorithm's feasibility was validated using a case study on PVC and Cu recovery from cable waste with a de-plasticiser and ball milling. The model was successfully trained based on limited experimental data to optimise the parameters. Accurate prediction can be achieved when predicting the dynamic behaviour of the separation, resulting in an MSE of 0.0015–0.0145. Based on the sensitivity analysis of the input features, the charged weight of cables and the impact energy were identified as crucial variables affecting separation efficiency. Besides several improvable points for the RNN model, we suggested that the ML algorithm can support intelligent design and operation for material recovery from cable waste. In the future, our research will apply this strategy for scaling up the cable separation process and other complex waste recycling processes (e.g., mechanochemical treatment of halogen-containing waste and pyrolysis of solid waste). ML will definitely support process optimisation and mitigate the associated environmental impacts, which can be widely applied to explore the process mechanisms and identify critical factors for accelerating the development of waste recycling technologies.

## Methods

**Model framework**. Based on the model framework in Fig.1, 'Parallel' does not refer to the distributed calculation on different processors (e.g., cores of CPU and GPU), but to the simultaneous model training of all datasets; 'asynchronous' refers to the various timesteps (advance of treatment time) during a calculation epoch. For one calculation step, the input features of a dataset include variables of process design and the separation yield ($Y_{sep}$) from the previous step (or initial zero). Furthermore, a new dimension representing the calculation timestep (d$t$ [s]) is created for $Y_{sep}$ of each dataset. Asynchronous timesteps are introduced because the model training will overfit the experimental results at the end of treatment based on our previous study[37]. With the dimension of d$t$, the parameters can be optimised to balance the prediction errors for the entire dynamic process.

With a three-dimensional (feature, dataset, and timestep) input tensor (*Input*), artificial neural networks are defined to predict a new $Y_{sep}$ with the advance of treatment time ($t$, [s]). In our previous ML modelling to predict a dechlorination reaction of PVC waste by ball milling[37], the output of each timestep was calculated by a pre-defined function based on the process mechanism[46]. However, for the cable separation by ball milling, only a qualitative mechanism was available[20]. Artificial neural networks can support the automatic formation of a flexible but robust relationship between the input and output without specified formulas to quantify the separation rate ($r_{sep}$ [s⁻¹]=d$Y_{sep}$/d$t$)[30,68]. Then, for each iteration in one timestep calculation, all the model parameters are updated using a backpropagation algorithm and the specified optimiser based on the error calculated by the predicted and experimental $Y_{sep}$ (loss)[69].

**Input features and data sources**. All the input features and data sources were reused from our previous study[20]. The input features consist of four process variables and simulated impact energy of the ball milling using the discrete element method. With the limited scope of experimental data, the grinding ball diameters and the cable densities, charged weights, and diameters are selected as the input features. The impact energy reveals the collision intensity between grind balls and cables during the ball milling process. The values of all input features are presented in Table 1.

The experimental $Y_{sep}$ versus treatment time under specific input features were also obtained from our previous study[20] as the training labels, which can be accessed with the code at github.com/wilsherelu/Cable-separation-prediction. However, for time series prediction by RNN, big data is commonly mandatory to train the model successfully[70,71]. Thus, we processed the reported experimental data into stepwise functions among the experimental $Y_{sep}$ over the treatment time. Despite the noise introduced by this data processing, the model was well trained based on the established RNN algorithm with the following deduction.

**Mathematical expression of specific RNN model**. In predicting the time series of $Y_{sep}$ under different treatment conditions, the most critical calculation is to quantify $r_{sep}$. Based on the proposed separation mechanism, Eq. 1 is assumed to establish a quantitative relationship between input features and $r_{sep}$:

$$r_{sep} = k \cdot (1 - c) \cdot (1 - Y_{sep}), \qquad (1)$$

where $k$ [s⁻¹] is a rate constant decided by specific input features; $c$ [-] is a constraint coefficient to control whether the cumulative collision between balls and cables is sufficient to separate PVC and Cu. Therefore, when $c = 1$, no separation occurs, while complete separation is achieved for $c = 0$ as an extension of the treatment time. The term $1-Y_{sep}$ is defined to exclude the separated cable sections. In this study, in accordance with $Y_{sep}$, $r_{sep}$, $k$ and $c$ are tensors with dataset and timestep dimensions. The next step is to model $k$ and $c$ based on the input features using Eqs. 2, 3

$$k\{dataset, dt\} = NN_k(Input\{dataset, feature, dt\}) \qquad (2)$$

$$c\{dataset, dt\} = NN_c(Input\{dataset, feature, dt\}) \qquad (3)$$

The detailed RNN architecture (input, hidden, and output layers, neuron numbers and activation functions) of $NN_k$ and $NN_c$ are included in Supplementary Notes 4 of SI. Given the calculated values of $k$ and $c$, the predicted $Y_{sep}$ at the next timestep can be derived by Eq. 4:

$$Y_{sep}(t + dt) = Y_{sep}(t) + r_{sep} \cdot dt \qquad (4)$$

Finally, the errors between the predicted $Y_{sep}$ and experimental data ($Y_{sep-exp}$) can be calculated by the customised loss function in Eq. 5, which weighs the absolute and relative errors simultaneously. The relative error is favourable for balancing the $Y_{sep}$ at different orders of magnitude; however, when $Y_{sep-exp}$ is low (<0.1%), a small prediction error will cause a considerable loss, which leads to overfitting for training periods with low $Y_{sep}$. Thus, the absolute error should be a major component based on trials with different combinations of weights.

$$Loss = 0.99\sum(Y_{sep-exp} - Y_{sep})^2 + 0.01\sum\left(\frac{Y_{sep-exp} - Y_{sep}}{Y_{sep-exp}}\right)^2 \qquad (5)$$

Model programming, training, and validation were accomplished using PyTorch 1.11.0, the most widely applied open source library for deep learning with an automatic differentiation system[72]. In this study, we did not modify the parameter optimisation framework of the default RNN model in PyTorch; instead, we just defined the aforementioned loss function and selected a widely applicable optimiser. We used the Adam method ($\beta_1 = 0.9$, $\beta_2 = 0.999$) for parameter optimisation based on the calculated loss[73].

## Data availability

Most of data supporting the findings of this study are available within the paper and and have been released alongside the code on https://github.com/wilsherelu/Cable-separation-prediction. For any remaining data, it is available from the corresponding author upon reasonable request.

## Code availability

The python code for the model can be found at https://github.com/wilsherelu/Cable-separation-prediction (https://doi.org/10.5281/zenodo.7811446).

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

## Acknowledgements

This research was supported by the Environment Research and Technology Development Fund [JPMEERF20223M02] of the Environmental Restoration and Conservation Agency of Japan and the Japan Society for the Promotion of Science KAKENHI grant [grant number 20H05708].

## Competing interests

The authors declare no competing interests.
