## [Peer review file · Communications Engineering]

Neural network based prediction of the efficacy of ball milling to separate cable waste materialsReviewers' comments:

Reviewer #1 (Remarks to the Author):

- 1) Don't mention the regression coeff values in the abstract or the conclusions.
- 2) Write info about the network training parameters in the abstract and how was the network configuration optimized?
- 3) Write the novelty of this work in 100 words (Intro).
- 4) The font size in all the figures and the graphical abstract is TOO SMALL.
- 5) For the industrial application of lab-scale recycling technologies, give at least 5 examples with details in the intro section.
- 6) Compare all the previous works done on this topic, together with their conditions, and results in the form of a NEW TABLE. Refer to REF from the last 5 years only.
- 7) In the results and discussions:
 - a) Discuss the individual and combined roles of grinding ball diameters and the cable densities, charged weights, and diameters with recent references. This has to be discussed in DETAIL.
 - b) What is the procedure for dynamic updating of the model parameters? How is it done and what are the quality control measures - DISCUSS IN 200 words with REFS.
 - c) When there are multiple figures within a figure, mention a, b, c, etc and provide captions.
 - d) How can the periods with higher uncertainty be resolved? Give your own perspectives in 100 words.
 - e) How can the noise from experimental fluctuation be avoided in future work? Give your perspectives.
 - f) Compare the optimal ball milling design for the high-quality recovery of the PVC and Cu with literature support, in ~ 200 words.
 - g) What are the other activation functions? Give examples.
- 8) Before the conclusions = Write the practical applications of this work in ~ 300 words, with REF support.
- 9) CHECK REF formatting manually. A lot of mistakes can be found.
- 10) All the comments from the reviewer should be incorporated in the revised MS.
- 11) The authors' response file should show all the new text added, new tables, and modified figures.
- 12) If the author's response file only mentions - DONE, OK, Thanks, ETC = then, the MS can be rejected.
- 13) A VERY GOOD work and congratulations to those who did this ANN simulation - but, I wish to see that the authors* (* I mean, all the authors of this paper) really read the comments, re-work and submit an MS that is revised completely.

Reviewer #2 (Remarks to the Author):

Thank you for allowing me to review this manuscript. The manuscript titled "Predicting the separation

of copper and poly(vinyl chloride) components from cable waste based on an asynchronous-parallel recurrent neural network " discusses a method for determining the optimal conditions for the process of separation of PVC from copper in waste electrical wires.

The manuscript was prepared with great care. The layout of the article is very clear. The descriptions of the individual chapters have been prepared very carefully.

I only have a few comments.

Please correct the spelling of poly(vinyl chloride).

Please comment on possible rescaling of the process. In the experiment, 12 g of cables were separated in 50 minutes - how big would an industrial device have to be to achieve satisfactory technical usability?

Is it possible to compare the calculated values in section 3.5 vv.336-344 to other separation methods? The reported results seem to be very large and unsatisfactory for industrial applications. What is the energy and CO₂-e benefit gained from the obtained recycled raw materials after separation? Providing this information without comment may spread false information about the lack of benefits of PVC recycling.

I believe that the manuscript can be published with minor corrections. The level of the article is adequate to the current rank of the journal.

Reviewer #3 (Remarks to the Author):

The paper introduces an RNN-based solution to predict the behavior of a ball-milling cable waste recycling separation process. The proposed approach was tested on a data set coming from a previous study of the authors.

Overall, the paper presents an interesting application of RNN to address a real world problem. However, important information to replicate the study is missing (please, see below), the experimental study only allows to validate the feasibility of the proposed solution (i.e., the approach is not compared to other state-of-the-art methods), and I was not able to grasp some statements, e.g., why is the method "asynchronous-parallel".

Detailed comments:

1. Why do you mean with "asynchronous-parallel". I may have not grasped the idea correctly, but I was not able to find the way you parallelized the RNN. Are you referring to a parallel (or distributed) training? How do you synchronize the network parameters? Please, comment on this.
2. Even though you included a reference to the original data set, it would be nice to have a description of the data, e.g., the number of samples, some descriptive stats of the data set, etc.
3. Describe the RNN architecture, otherwise it is not possible to replicate the study.
4. Would it be possible to release the implementation (code)?
5. How did you decide for the balance (i.e., the value of the parameters 0.99 and 0.01) in Eq. 5? What happens if those parameters are set with different values?
6. How did you set the Adam parameters?
7. You should compare the proposed method against state-of-the-art competitors. Otherwise, it is not possible to tell if the approach is "good or bad".

Reviewer#1

We sincerely appreciate your kind evaluation and valuable comments on our paper. We have carefully revised the manuscript according to your comments and suggestions. The modified contents have been highlighted in the manuscript. Please find our point-by-point responses to your comments below:

1. 'Don't mention the regression coeff values in the abstract or the conclusions.'

Reply:

Thanks for your suggestion. We have removed the coefficient of determination in the abstract and conclusion.

The text in Line 24-26 in page 2 was revised as '**High accuracy was confirmed based on a mean-square error of 0.0015–0.0145 of mean-square error between the prediction and experimental results.**'

We also done the removal in Line 411-413 in page 23-24 as '**Accurate prediction can be achieved when predicting the dynamic behaviour of the separation, resulting in an MSE of 0.0015–0.0145.**'

2. 'Write info about the network training parameters in the abstract and how was the network configuration optimized?'

Reply:

Thank you for your vital comment. The total number of epochs are 3600, which means the timestep is 1 second for one-hour treatment. For each epoch, the iteration times are 5 with a 0.0005 of learning rate. The design of the network was mainly based on the treatment mechanism. We tried different activation functions and hidden layer structures to improve the convergence performance of the model. The neuron numbers were set based on the size of input features.

We added this information to the abstract as '**First, with the input features, the model was successfully trained with six datasets (treatment conditions) at 3600 epochs; therefore, the timestep was 1 s for a one-hour treatment. One epoch contained five iteration and the learning rate was 0.0005.**' in Line 22-24 in Page 2.

3. 'Write the novelty of this work in 100 words (Intro).'

Reply:

Thanks for your constructive suggestion on adding the novelty in the introduction. To clearly addressing the novelty, we add a paragraph in the introduction as following.

'In this study, a modified recurrent neural network (RNN) was developed to model and predict the dynamics of the separation. ML was not directly used to predict the final separation yield based on the input features (experimental conditions) and treatment duration, but to predict the separation rate under various ball milling conditions during a short timeframe. Because the basic RNN method carries out parameter updating based on the real-time prediction loss, to avoid overfitting based on the end stage of treatment, different calculation timesteps were introduced in our model. Thus, training and predictions were performed simultaneously for all datasets (different experimental conditions and results) with different timesteps, which is known as parallel and asynchronous learning.' Was added in Line 87-95 in Page 6.

4. 'The font size in all the figures and the graphical abstract is TOO SMALL.'

Reply:

Sorry for the small text in the figures. We have increased the font size in all figures.

5. 'For the industrial application of lab-scale recycling technologies, give at least 5 examples with details in the intro section.'

Reply:

We agreed with your opinion about the industrial application potential. Mainly, we developed this process for cable waste with PVC cover, which can be widely found in e-waste and automobile wastes. However, with choosing a suitable solvent, it can be extended other types of cable waste. Thus, we have summarized these examples in the introduction.

We added '**This process has potential applications in the recycling of PVC cable waste from household electrical appliances, such as televisions and computers²³. With appropriate pre-treatment, this could also be applied to cable mixed in construction²⁴ and automobile²⁵ waste. In terms of high-voltage transmission cables, this process could be modified with a different solvent because the covers are commonly made of polyethylene²⁶.**' Was added in Line 65-70 in Page 5.

6. 'Compare all the previous works done on this topic, together with their conditions, and results in the form of a NEW TABLE. Refer to REF from the last 5 years only.'

Reply:

Although this paper is not a review paper, we think your advice is valuable to improve the quality of this paper. Thus, except our group's researches, we summarized all papers related with waste cable separation in the last 5 years based on the Web of Science database in Table S1 of supporting information.

We added '**Based on the literature review summarised in Table S1 of SI, many efficient processes for cable separation have been developed at lab scale; however, modelling the process was rare, especially for ML.**' Was added in Line 73-75 in Page 5.

7. 'In the results and discussions:

a) Discuss the individual and combined roles of grinding ball diameters and the cable densities, charged weights, and diameters with recent references. This has to be discussed in DETAIL.'

Reply:

We also thought the current discussion on the influence of ball and cable properties is insufficient. However, there is few literatures about the separation mechanism of cable by ball milling. Thus, we referred to fundamental analysis of ball milling process by discrete element method.

We added '**This phenomenon can also be supported by the discrete element method simulation on the different charging ball mill ratio**⁴⁴.' in Line 279-280 in Page 17;

'In principle, a lower density will cause a decreased diameter for a certain weight and length of cables, and it has been reported that small particles have less contact chance compared with large ones⁴⁵.' in Line 281-283 in Page 17;

'The increase in ball diameter is not equivalent to the enhanced mechanical efficiency of ball milling⁴⁶.' in Line 297-298 in Page 18-19;

'Because the charged cable weight and impact energy are critical factors for separation, the increase in impact energy (e.g., by increasing the rotation speed) should be accompanied when raising the throughput.' in Line 348-350 in Page 21;

8. 'What is the procedure for dynamic updating of the model parameters? How is it done and what are the quality control measures - DISCUSS IN 200 words with REFS.'

Reply:

Thanks for your question on the updating of the model parameters. In this paper, we did not modify the parameter updating framework of the basic RNN model in pytorch. And the discussion of detailed RNN algorithm is out of scope for this study. To make this point clearer, we emphasized the statement in the method part.

We revised the text in Line 130-132 in Page 8 as '**Then, for each iteration in one timestep calculation, all the model parameters are updated using a backpropagation algorithm and the specified optimiser based on the error calculated by the predicted and experimental Y_{sep} (loss)³⁷.**';

We added '**In this study, we did not modify the parameter optimization framework of the default RNN model in PyTorch; instead, we just defined the aforementioned loss function and selected a widely applicable optimiser.**' in Line 186-188 in Page 11.

9. 'When there are multiple figures within a figure, mention a, b, c, etc and provide captions.'

Reply:

We sorry for the missing caption of subplots in Fig. 2 and 3. We have fixed this issue in the revised paper.

We the caption of in Fig. 2 in Line 205-207 in Page 13 as '**Visualisation of the training process to optimise model parameters: (a)–(c) varying cable diameter and weight with fixed 15-mm ball (entries 1-3); (d)–(e) varying cable diameter and weight with fixed 20-mm ball (entries 4-6).**';

We the caption of in Fig. 3 in Line 252-255 in Page 16 as '**Comparison between $Y_{sep-exp}$ and the predicted Y_{sep} for all experimental conditions based on the trained model with optimised parameters: (a)–(c) varying cable diameter and weight with fixed 15-mm ball (entries 1-3); (d)–(e) varying cable diameter and weight with fixed 20-mm ball (entries 4-6).**'.

10. 'How can the periods with higher uncertainty be resolved? Give your own perspectives in 100 words.'

Reply:

Regarding the methods to reduce the uncertainty of training process, we think the solutions can be mainly realized based on the experimental efforts. For example, appropriate increase of sampling interval can form a smoother training label for each dataset. In consequence, the separation rate will not result in a sudden change. In addition, some random factors are existent during the ball milling process. Thus, a larger ball mill can reduce the uncertainty from the collision between balls and cables.

We the caption of in Fig. 2 in Line 233-237 in Page 14 as '**To reduce this uncertainty, a better sampling-interval design can be suggested. For example, increasing the data points around the starting point of the separation when both the Y_{sep} and r_{sep} change sharply. Meanwhile, experiments with a scaled-up ball mill also reduce the randomness of ball-to-cable collision.**'.

11. ‘How can the noise from experimental fluctuation be avoided in future work? Give your perspectives.’

Reply:

The noise from experimental fluctuation is an evitable problem for the modeling study. Especially, it is hard to control the quality of parallel experiments because the real waste sample has high heterogeneity. However, for the industrial application, the prediction accuracy is not the priority. It is more important to derive a robust solution which can efficiently handle mixed cables with various diameters and other properties. Thus, we concern about that the experimental noise will not affect the convergence ability of the model. We propose that the dropout method can help overcome the noise from experimental data and avoid overfitting.

We added ‘**To avoid the influence of such noise and experimental errors on the training, a dropout method which randomly omits neurons in hidden layers can be adopted to mitigate overfitting**⁴³.’ in Line 223-225 in Page 14.

12. ‘Compare the optimal ball milling design for the high-quality recovery of the PVC and Cu with literature support, in ~ 200 words.’

Reply:

Thank you for the comment on the process optimization. Our ultimate purpose of establishing this RNN model is to realize the process optimization for industrial application. However, this paper is only the first step. As you can find in Fig. 4, we currently could find a local-optimal process variable for achieve the highest separation yield under a certain treatment time and cable property. While, for suggesting the optimal process design, a lot of additional work is necessary. At first, this study only considered one process; however, we have developed several alternatives to recycle waste cables. Thus, the applicability of this model to other similar processes should be validated in future. On the other hand, for industrial application, scaling up of the process should be also considered in the future. Finally, the separation yield is not the only optimization objection. Separation accuracy and environmental impacts (e.g., carbon footprint) are more emphasized in our project. Thus, we added such discussion in the paper.

We added ‘**The non-linear model can be optimised to suggest an energy-saving, environmentally-friendly, economically-feasible design for emerging recycling technologies**⁴⁸, **which is the ultimate target of our research. However, for the current result shown in Fig. 4, only the local-optimal solution can be obtained for the maximum Y_{sep} with a fixed treatment time and cable type. In the future, the applicability of the developed model for different process designs and ball milling scales should be validated.**’ in Line 325-331 in Page 20.

We added 'Because the charged cable weight and impact energy are critical factors for separation, the increase in impact energy (e.g., by increasing the rotation speed) should be accompanied when raising the throughput. A moderate rotation speed may also exist to maximise the ball milling efficiency ⁵². For various cable specifications, the optimal mechanical conditions differ in terms of separation efficiency. In general, the optimal conditions vary based on the sample properties ^{35,53}.' in Line 348-353 in Page 21.

13. 'What are the other activation functions? Give examples.'

Reply:

Some examples were added based on your kind comment.

'The complexity of the model during construction can be enhanced by pretreating the input features (e.g., applying the power function), increasing the hidden layers, and introducing other activation functions (e.g., ReLU, tanh, or their combination ⁵¹).' has been revised in Line 340-342 in Page 20.

14. 'Before the conclusions = Write the practical applications of this work in ~ 300 words, with REF support.'

Reply:

We appreciate your valuable suggestion. Currently, there is a section '3.5 Prospect of industrial application' before the conclusion part. We hope you didn't mis-download the initial version of our manuscript without this part. However, this part more focused on the analysis of potential environmental benefits after the industrial application of this work. Thus, we additionally extend the prospect to other aspects.

'Overall, the developed RNN model can be applied to other ball milling processes, even to general industrial processes affected by dynamic factors. For example, ball milling can facilitate the mechanochemical treatment of solid waste^{64, 65, 66} which commonly operates as a batch reactor. The RNN model can be adapted to predict chemical reaction rates under various temperatures, reactant concentrations, and other parameters. Moreover, ML has been widely applied in computational fluid dynamics to improve simulation efficiency and accuracy^{67, 68}. The multi-timestep training method proposed in this study has the potential to be applied in such algorithms to improve parameter optimisation.' has been revised in Line 398-405 in Page 23.

15. 'CHECK REF formatting manually. A lot of mistakes can be found.'

Reply:

Thank you for your reminder. We have carefully checked the bibliography. You are right that there were some mistakes. We have revised the wrong abbreviation format of journal names. Some specific errors were also fixed, such as the access date of web pages, the conference name of proceeding papers, and the publisher of books.

16. 'All the comments from the reviewer should be incorporated in the revised MS.'

Reply:

Yes, this response letter is integrated for all reviewers.

17. 'The authors' response file should show all the new text added, new tables, and modified figures.'

Reply:

Based on your kind comment, we have addressed all revisions in this letter.

17. 'If the author's response file only mentions - DONE, OK, Thanks, ETC = then, the MS can be rejected.'

Reply:

Of course, we would not make these informal replies.

17. 'A VERY GOOD work and congratulations to those who did this ANN simulation - but, I wish to see that the authors* (* I mean, all the authors of this paper) really read the comments, re-work and submit an MS that is revised completely.'

Reply:

Thanks so much for your conclusive and positive comment. The revised paper and this response letter have been approved by all authors.

Reviewer#2

1. 'Thank you for allowing me to review this manuscript. The manuscript titled "Predicting the separation of copper and poly(vinyl chloride) components from cable waste based on an asynchronous-parallel recurrent neural network " discusses a method for determining the optimal conditions for the process of separation of PVC from copper in waste electrical wires.

The manuscript was prepared with great care. The layout of the article is very clear. The descriptions of the individual chapters have been prepared very carefully.

I only have a few comments.'

Reply:

We sincerely appreciate your positive evaluation and precious comments on our papers. The manuscript has been furtherly updated according to your suggestions.

2. 'Please correct the spelling of poly(vinyl chloride).'

Reply:

Thanks for your suggestion. We replaced all 'polyvinyl chloride' with 'poly(vinyl chloride)' in the title, abstract, and main text.

3. 'Please comment on possible rescaling of the process. In the experiment, 12 g of cables were separated in 50 minutes - how big would an industrial device have to be to achieve satisfactory technical usability?'

Reply:

In terms of scale problem, it is essential to be considered in our future study. During the experimental work, we didn't set a large filling ratio of ball mill. In other words, there is a considerable potential to increase the throughput. Based on another project about the dechlorination of PVC by ball milling (*ACS Sustainable Chem. Eng.* 2021, 9, 42, 14112–14123), 1 kg PVC waste can be treated in a ball mill with 26-cm diameter and 60-cm length (filling ratio was about 50% including solvent and balls). In addition, with the scaling up of ball mill, the increase of impact energy can shorten the treatment time.

Moreover, we added '**In the future, the applicability of the developed model for different process designs and ball milling scales should be validated.**' in Line 329-331 in Page 20.

3. 'Is it possible to compare the calculated values in section 3.5 vv.336-344 to other separation methods? The reported results seem to be very large and unsatisfactory for industrial applications. What is the energy and CO₂-e benefit gained from the obtained recycled raw materials after separation? Providing this information without comment may spread false information about the lack of benefits of PVC recycling.'

Reply:

Thanks so much for your comment. As the life cycle assessment is not the main scope of this study, we don't compare our process with other processes. In addition, the estimated results in Section 3.5 are rough because it is based on lab-scale experiments. During the scaling up, the efficiency should be increased a lot. Thus, the energy consumption will be reduced. And we supposed you misunderstood some points. Thus, we revised some statements to avoid the potential ambiguity.

We revised the text '**For a bench-scale ball mill reactor, a 50-min treatment is necessary for achieving complete separation of 65-g cable waste ²², in which the throughput and energy efficiency are not optimised. Assuming the operating power of ball milling as 100 W ⁶³, the electricity consumption is 1.28 kWh/kg separated cable waste, resulting in a 0.85 kg CO₂-e/kg separated cable waste of GHG emissions based on the current power grid of Japan. Thus, even though the lab-scale treatment efficiency is low, the avoided GHG emissions would be far higher than the additionally produced ones.**' in Line 388-394 in Page 22-23.

4. 'I believe that the manuscript can be published with minor corrections. The level of the article is adequate to the current rank of the journal.'

Reply:

It is our great honor to receive your highly positive recommendation.

Reviewer#3

1. ‘ The paper introduces an RNN-based solution to predict the behavior of a ball-milling cable waste recycling separation process. The proposed approach was tested on a data set coming from a previous study of the authors.

Overall, the paper presents an interesting application of RNN to address a real world problem. However, important information to replicate the study is missing (please, see below), the experimental study only allows to validate the feasibility of the proposed solution (i.e., the approach is not compared to other state-of-the-art methods), and I was not able to grasp some statements, e.g., why is the method “asynchronous-parallel”.’

Reply:

Thank you so much for the positive summary and kind comments on our paper. We agreed with your opinion that the replication of research is vital in academic. We also realized some statements are not so clear in the original submission. Thus, we modified the paper based on your valuable suggestions to improve the quality.

2. 'Why do you mean with "asynchronous-parallel". I may have not grasped the idea correctly, but I was not able to find the way you parallelized the RNN. Are you referring to a parallel (or distributed) training? How do you synchronize the network parameters? Please, comment on this.'

Reply:

We are sorry for unclear definition of the model name. We supposed that you tried understanding the asynchronous and parallel based on the concept of program execution in computer science. However, we defined two terms for interpreting the innovation on training process. The parallel means that all datasets are simultaneously trained for obtaining universe parameters, rather than the simultaneous program execution among several threads or hardware (CPUs, GPUs). On the other hand, asynchronous means that various calculation timesteps for representing the advanced treatment time are existent during one epoch. This is designed for optimizing the parameters with covering different periods in a time series. To clarify these definitions, we made the following revisions.

The text in Line 91-95 Page 6 as '**Because the basic RNN method carries out parameter updating based on the real-time prediction loss, to avoid overfitting based on the end stage of treatment, different calculation timesteps were introduced in our model. Thus, training and predictions were performed simultaneously for all datasets (different experimental conditions and results) with different timesteps, which is known as parallel and asynchronous learning.**';

'Parallel' does not refer to the distributed calculation on different processors (e.g., cores of CPU and GPU), but to the simultaneous model training of all

datasets; ‘asynchronous’ refers to the various timesteps (advance of treatment time) during a calculation epoch.’ in Line 112-114 Page 7 was revised.

3. ‘Even though you included a reference to the original data set, it would be nice to have a description of the data, e.g., the number of samples, some descriptive stats of the data set, etc.’

Reply:

Thank you very much for your constructive suggestion. Data declaration is important for replicating our work. Thus, we uploaded the excel files of original data on the GitHub with the code.

The text in Line 145-147 Page 9 as **‘The experimental Y_{sep} versus treatment time under specific input features were also obtained from our previous study ²¹ as the training labels, which can be accessed with the code at github.com/wilsherlu/Cable-separation-prediction.’**

4. 'Describe the RNN architecture, otherwise it is not possible to replicate the study.'

Reply:

We are sorry about the unclear guide of the RNN architecture. It was included in S1.3 in supporting information. In addition, the open-access code can help understand the RNN architecture.

The text in Line 171-172 Page 10 as '**The detailed RNN architecture (input, hidden, and output layers, neuron numbers, and activation functions) of NN_k and NN_e are included in Section S3 of SI.**'

5. 'Would it be possible to release the implementation (code)?'

Reply:

Yes. The original code has been released at GitHub (<https://github.com/wilshere/lu/Cable-separation-prediction>).

6. ‘How did you decide for the balance (i.e., the value of the parameters 0.99 and 0.01) in Eq. 5? What happens if those parameters are set with different values?’

Reply:

Commonly, the absolute error is applied in the loss function. While, we intended to increase the weight for the training labels whose R_{sep} is zero. We have tried different combination of parameters. If the weight of relative error is larger (e.g., 0.05), the overfitting problem will also occur so that it is hard to update the parameters when the separation stage starts.

The text in Line 178-182 Page 10-11 as ‘**The relative error is favourable for balancing the Y_{sep} at different orders of magnitude; however, when $Y_{\text{sep-exp}}$ is low (less than 0.1%), a small prediction error will cause a significant loss, which leads to overfitting for training periods with low Y_{sep} . Thus, the absolute error should be a major component based on trials with different combinations of weights.**’.

7. ‘How did you set the Adam parameters?’

Reply:

We applied the default parameter values ($\beta_1 = 0.9$, $\beta_2 = 0.999$) of Adam optimizer. Since we found it is capable of carrying out successful prediction, we did not modify these parameters.

8. 'You should compare the proposed method against state-of-the-art competitors.

Otherwise, it is not possible to tell if the approach is "good or bad".'

Reply:

Thank you very much for your vital suggestion. The machine learning study on the prediction of cable separation is scarce. In addition, we realize the prediction for the dynamic behavior of cable separation. Thus, we strengthened this point in the paper.

We added '**Based on the literature review summarised in Table S1 of SI, many efficient processes for cable separation have been developed at lab scale; however, modelling the process was rare, especially for ML.**' in Line 73-76 Page 5;

And, '**There are existing studies which model cable separation processes as polynomial ⁴⁹ or support vector regression ⁵⁰; however, none of these works carry out dynamic modelling of the process, which is innovatively realised by RNN.**' in Line 332-334 Page 20.

Reviewer#1

We sincerely appreciate your kind evaluation and valuable comments on our paper. We have carefully revised the manuscript according to your comments and suggestions. The modified contents have been highlighted in the manuscript. Please find our point-by-point responses to your comments below:

1. 'Don't mention the regression coeff values in the abstract or the conclusions.'

Reply:

Thanks for your suggestion. We have removed the coefficient of determination in the abstract and conclusion.

The text in Line 24-26 in page 2 was revised as '**High accuracy was confirmed based on a mean-square error of 0.0015–0.0145 of mean-square error between the prediction and experimental results.**'

We also done the removal in Line 435-437 in page 25 as '**Accurate prediction can be achieved when predicting the dynamic behaviour of the separation, resulting in an MSE of 0.0015–0.0145.**'

2. 'Write info about the network training parameters in the abstract and how was the network configuration optimized?'

Reply:

Thank you for your vital comment. The total number of epochs are 3600, which means the timestep is 1 second for one-hour treatment. For each epoch, the iteration times are 5 with a 0.0005 of learning rate. The design of the network was mainly based on the treatment mechanism. We tried different activation functions and hidden layer structures to improve the convergence performance of the model. The neuron numbers were set based on the size of input features.

We added this information to the abstract as '**First, with the input features, the model was successfully trained with six datasets (treatment conditions) at 3600 epochs; therefore, the timestep was 1 s for a one-hour treatment. One epoch contained five iteration and the learning rate was 0.0005.**' in Line 22-24 in Page 2.

3. 'Write the novelty of this work in 100 words (Intro).'

Reply:

Thanks for your constructive suggestion on adding the novelty in the introduction.

To clearly addressing the novelty, we add a paragraph in the introduction as following.

'In this study, a modified RNN was developed to model and predict the dynamics of the separation. ML was not directly used to predict the final separation yield based on the input features (experimental conditions) and treatment duration, but to predict the separation rate under various ball milling conditions during a short timeframe. Because the basic RNN method carries out parameter updating based on the real-time prediction loss, to avoid overfitting based on the end stage of treatment, different calculation timesteps were introduced in our model. Thus, training and predictions were performed simultaneously for all datasets (different experimental conditions and results) with different timesteps, which is known as parallel and asynchronous learning.' was

added in Line 101-109 in Page 6-7.

4. 'The font size in all the figures and the graphical abstract is TOO SMALL.'

Reply:

Sorry for the small text in the figures. We have increased the font size in all figures.

5. 'For the industrial application of lab-scale recycling technologies, give at least 5 examples with details in the intro section.'

Reply:

We agreed with your opinion about the industrial application potential. Mainly, we developed this process for cable waste with PVC cover, which can be widely found in e-waste and automobile wastes. However, with choosing a suitable solvent, it can be extended other types of cable waste. Thus, we have summarized these examples in the introduction.

We added '**This process has potential applications in the recycling of PVC cable waste from household electrical appliances, such as televisions and computers²⁵. With appropriate pre-treatment, this could also be applied to cable mixed in construction²⁶ and automobile²⁷ waste. In terms of high-voltage transmission cables, this process could be modified with a different solvent because the covers are commonly made of polyethylene²⁸.**' Was added in Line 72-76 in Page 5.

6. 'Compare all the previous works done on this topic, together with their conditions, and results in the form of a NEW TABLE. Refer to REF from the last 5 years only.'

Reply:

Although this paper is not a review paper, we think your advice is valuable to improve the quality of this paper. Thus, except our group's researches, we summarized all papers related with waste cable separation in the last 5 years based on the Web of Science database in Table S1 of supporting information.

We added '**Based on the literature review summarised in Table S1 of SI, many efficient processes for cable separation have been developed at a lab scale; however, process modelling was rare, especially for yield prediction by ML.**' was added in Line 79-82 in Page 5.

7. 'In the results and discussions:

a) Discuss the individual and combined roles of grinding ball diameters and the cable densities, charged weights, and diameters with recent references. This has to be discussed in DETAIL.'

Reply:

We also thought the current discussion on the influence of ball and cable properties is insufficient. However, there is few literatures about the separation mechanism of cable by ball milling. Thus, we referred to fundamental analysis of ball milling process by discrete element method.

We added '**This phenomenon can also be supported by the discrete element method simulation on the different charging ball mill ratio ⁴⁸.**' in Line 295-296 in Page 18;

'In principle, a lower density will cause a decreased diameter for a certain weight and length of cables, and it has been reported that small particles have less contact chance compared with large ones ⁴⁹.' in Line 297-299 in Page 18;

'The increase in ball diameter is not equivalent to the enhanced mechanical efficiency of ball milling ⁵⁰.' in Line 313-314 in Page 19-20;

'Because the charged cable weight and impact energy are critical factors for separation, the increase in impact energy (e.g., by increasing the rotation speed) should be accompanied when raising the throughput.'' in Line 361-363 in Page 22;

8. 'What is the procedure for dynamic updating of the model parameters? How is it done and what are the quality control measures - DISCUSS IN 200 words with REFS.'

Reply:

Thanks for your question on the updating of the model parameters. In this paper, we did not modify the parameter updating framework of the basic RNN model in pytorch. And the discussion of detailed RNN algorithm is out of scope for this study. To make this point clearer, we emphasized the statement in the method part.

We revised the text in Line 144-146 in Page 8 as '**Then, for each iteration in one timestep calculation, all the model parameters are updated using a backpropagation algorithm and the specified optimiser based on the error calculated by the predicted and experimental Y_{sep} (loss) ⁴¹.**';

We added '**In this study, we did not modify the parameter optimization framework of the default RNN model in PyTorch; instead, we just defined the aforementioned loss function and selected a widely applicable optimiser.**' in Line 200-202 in Page 11-12.

9. 'When there are multiple figures within a figure, mention a, b, c, etc and provide captions.'

Reply:

We sorry for the missing caption of subplots in Fig. 2 and 3. We have fixed this issue in the revised paper.

We the caption of in Fig. 2 in Line 221-223 in Page 14 as '**Visualisation of the training process to optimise model parameters: (a)–(c) varying cable diameter and weight with fixed 15-mm ball (entries 1-3); (d)–(e) varying cable diameter and weight with fixed 20-mm ball (entries 4-6).**';

We the caption of in Fig. 3 in Line 268-271 in Page 17 as '**Comparison between $Y_{sep-exp}$ and the predicted Y_{sep} for all experimental conditions based on the trained model with optimised parameters: (a)–(c) varying cable diameter and weight with fixed 15-mm ball (entries 1-3); (d)–(e) varying cable diameter and weight with fixed 20-mm ball (entries 4-6).**'.

10. 'How can the periods with higher uncertainty be resolved? Give your own perspectives in 100 words.'

Reply:

Regarding the methods to reduce the uncertainty of training process, we think the solutions can be mainly realized based on the experimental efforts. For example, appropriate increase of sampling interval can form a smoother training label for each dataset. In consequence, the separation rate will not result in a sudden change. In addition, some random factors are existent during the ball milling process. Thus, a larger ball mill can reduce the uncertainty from the collision between balls and cables.

We added in Line 249-253 in Page 15-16 as '**To reduce this uncertainty, a better sampling-interval design can be suggested. For example, increasing the data points around the starting point of the separation when both the Y_{sep} and r_{sep} change sharply. Meanwhile, experiments with a scaled-up ball mill also reduce the randomness of ball-to-cable collision.**'.

11. ‘How can the noise from experimental fluctuation be avoided in future work? Give your perspectives.’

Reply:

The noise from experimental fluctuation is an evitable problem for the modeling study. Especially, it is hard to control the quality of parallel experiments because the real waste sample has high heterogeneity. However, for the industrial application, the prediction accuracy is not the priority. It is more important to derive a robust solution which can efficiently handle mixed cables with various diameters and other properties. Thus, we concern about that the experimental noise will not affect the convergence ability of the model. We propose that the dropout method can help overcome the noise from experimental data and avoid overfitting.

We added ‘**To avoid the influence of such noise and experimental errors on the training, a dropout method which randomly omits neurons in hidden layers can be adopted to mitigate overfitting**⁴⁷.’ in Line 239-241 in Page 15.

12. ‘Compare the optimal ball milling design for the high-quality recovery of the PVC and Cu with literature support, in ~ 200 words.’

Reply:

Thank you for the comment on the process optimization. Our ultimate purpose of establishing this RNN model is to realize the process optimization for industrial application. However, this paper is only the first step. As you can find in Fig. 4, we currently could find a local-optimal process variable for achieve the highest separation yield under a certain treatment time and cable property. While, for suggesting the optimal process design, a lot of additional work is necessary. At first, this study only considered one process; however, we have developed several alternatives to recycle waste cables. Thus, the applicability of this model to other similar processes should be validated in future. On the other hand, for industrial application, scaling up of the process should be also considered in the future. Finally, the separation yield is not the only optimization objection. Separation accuracy and environmental impacts (e.g., carbon footprint) are more emphasized in our project. Thus, we added such discussion in the paper.

We added ‘**The non-linear model can be optimised to suggest an energy-saving, environmentally-friendly, economically-feasible design for emerging recycling technologies**⁴⁸, which is the ultimate target of our research. However, for the current result shown in Fig. 4, only the local-optimal solution can be obtained for the maximum Y_{sep} with a fixed treatment time and cable type. In the future, the applicability of the developed model for different process designs and ball milling scales should be validated.’ in Line 341-347 in Page 21.

We added 'Because the charged cable weight and impact energy are critical factors for separation, the increase in impact energy (e.g., by increasing the rotation speed) should be accompanied when raising the throughput. A moderate rotation speed may also exist to maximise the ball milling efficiency⁵⁴. For various cable specifications, the optimal mechanical conditions differ in terms of separation efficiency. In general, the optimal conditions vary based on the sample properties^{39,55}.' in Line 361-366 in Page 22.

13. 'What are the other activation functions? Give examples.'

Reply:

Some examples were added based on your kind comment.

'The complexity of the model during construction can be enhanced by pretreating the input features (e.g., applying the power function), increasing the hidden layers, and introducing other activation functions (e.g., ReLU, tanh, or their combination⁵³).' has been revised in Line 353-356 in Page 21.

14. 'Before the conclusions = Write the practical applications of this work in ~ 300 words, with REF support.'

Reply:

We appreciate your valuable suggestion. Currently, there is a section '3.5 Prospect of industrial application' before the conclusion part. We hope you didn't mis-download the initial version of our manuscript without this part. However, this part more focused on the analysis of potential environmental benefits after the industrial application of this work. Thus, we additionally extend the prospect to other aspects.

'Overall, the developed RNN model can be applied to other ball milling processes, even to general industrial processes affected by dynamic factors. For example, ball milling can facilitate the mechanochemical treatment of solid waste ^{70, 71, 72} which commonly operates as a batch reactor. The RNN model can be adapted to predict chemical reaction rates under various temperatures, reactant concentrations, and other parameters. Moreover, ML has been widely applied in computational fluid dynamics to improve simulation efficiency and accuracy ^{73, 74}. The multi-timestep training method proposed in this study has the potential to be applied in such algorithms to improve parameter optimisation.' has been revised in Line 422-429 in Page 24-25.

15. 'CHECK REF formatting manually. A lot of mistakes can be found.'

Reply:

Thank you for your reminder. We have carefully checked the bibliography. You are right that there were some mistakes. We have revised the wrong abbreviation format of journal names. Some specific errors were also fixed, such as the access date of web pages, the conference name of proceeding papers, and the publisher of books.

16. 'All the comments from the reviewer should be incorporated in the revised MS.'

Reply:

Yes, this response letter is integrated for all reviewers.

17. 'The authors' response file should show all the new text added, new tables, and modified figures.'

Reply:

Based on your kind comment, we have addressed all revisions in this letter.

17. 'If the author's response file only mentions - DONE, OK, Thanks, ETC = then, the MS can be rejected.'

Reply:

Of course, we would not make these informal replies.

17. 'A VERY GOOD work and congratulations to those who did this ANN simulation - but, I wish to see that the authors* (* I mean, all the authors of this paper) really read the comments, re-work and submit an MS that is revised completely.'

Reply:

Thanks so much for your conclusive and positive comment. The revised paper and this response letter have been approved by all authors.

Reviewer#2

1. 'Thank you for allowing me to review this manuscript. The manuscript titled "Predicting the separation of copper and poly(vinyl chloride) components from cable waste based on an asynchronous-parallel recurrent neural network " discusses a method for determining the optimal conditions for the process of separation of PVC from copper in waste electrical wires.

The manuscript was prepared with great care. The layout of the article is very clear. The descriptions of the individual chapters have been prepared very carefully.

I only have a few comments.'

Reply:

We sincerely appreciate your positive evaluation and precious comments on our papers. The manuscript has been furtherly updated according to your suggestions.

2. 'Please correct the spelling of poly(vinyl chloride).'

Reply:

Thanks for your suggestion. We replaced all 'polyvinyl chloride' with 'poly(vinyl chloride)' in the title, abstract, and main text.

3. 'Please comment on possible rescaling of the process. In the experiment, 12 g of cables were separated in 50 minutes - how big would an industrial device have to be to achieve satisfactory technical usability?'

Reply:

In terms of scale problem, it is essential to be considered in our future study. During the experimental work, we didn't set a large filling ratio of ball mill. In other words, there is a considerable potential to increase the throughput. Based on another project about the dechlorination of PVC by ball milling (*ACS Sustainable Chem. Eng.* 2021, 9, 42, 14112–14123), 1 kg PVC waste can be treated in a ball mill with 26-cm diameter and 60-cm length (filling ratio was about 50% including solvent and balls). In addition, with the scaling up of ball mill, the increase of impact energy can shorten the treatment time.

Moreover, we added '**In the future, the applicability of the developed model for different process designs and ball milling scales should be validated.**' in Line 345-347 in Page 21.

3. 'Is it possible to compare the calculated values in section 3.5 vv.336-344 to other separation methods? The reported results seem to be very large and unsatisfactory for industrial applications. What is the energy and CO₂-e benefit gained from the obtained recycled raw materials after separation? Providing this information without comment may spread false information about the lack of benefits of PVC recycling.'

Reply:

Thanks so much for your comment. As the life cycle assessment is not the main scope of this study, we don't compare our process with other processes. In addition, the estimated results in Section 3.5 are rough because it is based on lab-scale experiments. During the scaling up, the efficiency should be increased a lot. Thus, the energy consumption will be reduced. And we supposed you misunderstood some points. Thus, we revised some statements to avoid the potential ambiguity.

We revised the text '**For a bench-scale ball mill reactor, a 50-min treatment is necessary for achieving complete separation of 65-g cable waste²⁴, in which the throughput and energy efficiency are not optimised. Assuming the operating power of ball milling as 100 W⁶⁵, the electricity consumption is 1.28 kWh/kg separated cable waste, resulting in a 0.85 kg CO₂-e/kg separated cable waste of GHG emissions based on the current power grid of Japan. Thus, even though the lab-scale treatment efficiency is low, the avoided GHG emissions would be far higher than the additionally produced ones.**' in Line 401-407 in Page 23-24.

4. 'I believe that the manuscript can be published with minor corrections. The level of the article is adequate to the current rank of the journal.'

Reply:

It is our great honor to receive your highly positive recommendation.

Reviewer#3

1. ‘ The paper introduces an RNN-based solution to predict the behavior of a ball-milling cable waste recycling separation process. The proposed approach was tested on a data set coming from a previous study of the authors.

Overall, the paper presents an interesting application of RNN to address a real world problem. However, important information to replicate the study is missing (please, see below), the experimental study only allows to validate the feasibility of the proposed solution (i.e., the approach is not compared to other state-of-the-art methods), and I was not able to grasp some statements, e.g., why is the method “asynchronous-parallel”.’

Reply:

Thank you so much for the positive summary and kind comments on our paper. We agreed with your opinion that the replication of research is vital in academic. We also realized some statements are not so clear in the original submission. Thus, we modified the paper based on your valuable suggestions to improve the quality.

2. 'Why do you mean with "asynchronous-parallel". I may have not grasped the idea correctly, but I was not able to find the way you parallelized the RNN. Are you referring to a parallel (or distributed) training? How do you synchronize the network parameters? Please, comment on this.'

Reply:

We are sorry for unclear definition of the model name. We supposed that you tried understanding the asynchronous and parallel based on the concept of program execution in computer science. However, we defined two terms for interpreting the innovation on training process. The parallel means that all datasets are simultaneously trained for obtaining universe parameters, rather than the simultaneous program execution among several threads or hardware (CPUs, GPUs). On the other hand, asynchronous means that various calculation timesteps for representing the advanced treatment time are existent during one epoch. This is designed for optimizing the parameters with covering different periods in a time series. To clarify these definitions, we made the following revisions.

The text in Line 104-109 Page 6-7 as **'Because the basic RNN method carries out parameter updating based on the real-time prediction loss, to avoid overfitting based on the end stage of treatment, different calculation timesteps were introduced in our model. Thus, training and predictions were performed simultaneously for all datasets (different experimental conditions and results) with different timesteps, which is known as parallel and asynchronous learning.'**;

'Parallel' does not refer to the distributed calculation on different processors (e.g., cores of CPU and GPU), but to the simultaneous model training of all

datasets; ‘asynchronous’ refers to the various timesteps (advance of treatment time) during a calculation epoch.’ in Line 126-128 Page 7-8 was revised.

3. ‘Even though you included a reference to the original data set, it would be nice to have a description of the data, e.g., the number of samples, some descriptive stats of the data set, etc.’

Reply:

Thank you very much for your constructive suggestion. Data declaration is important for replicating our work. Thus, we uploaded the excel files of original data on the GitHub with the code.

The text in Line 159-161 Page 9 as **‘The experimental Y_{sep} versus treatment time under specific input features were also obtained from our previous study ²¹ as the training labels, which can be accessed with the code at github.com/wilsherlu/Cable-separation-prediction.’**

4. 'Describe the RNN architecture, otherwise it is not possible to replicate the study.'

Reply:

We are sorry about the unclear guide of the RNN architecture. It was included in S1.3 in supporting information. In addition, the open-access code can help understand the RNN architecture.

The text in Line 185-186 Page 11 as '**The detailed RNN architecture (input, hidden, and output layers, neuron numbers, and activation functions) of NN_k and NN_e are included in Section S3 of SI.**'

5. 'Would it be possible to release the implementation (code)?'

Reply:

Yes. The original code has been released at GitHub (<https://github.com/wilshere/lu/Cable-separation-prediction>).

6. ‘How did you decide for the balance (i.e., the value of the parameters 0.99 and 0.01) in Eq. 5? What happens if those parameters are set with different values?’

Reply:

Commonly, the absolute error is applied in the loss function. While, we intended to increase the weight for the training labels whose R_{sep} is zero. We have tried different combination of parameters. If the weight of relative error is larger (e.g., 0.05), the overfitting problem will also occur so that it is hard to update the parameters when the separation stage starts.

The text in Line 192-196 Page 11 as **‘The relative error is favourable for balancing the Y_{sep} at different orders of magnitude; however, when $Y_{\text{sep-exp}}$ is low (less than 0.1%), a small prediction error will cause a significant loss, which leads to overfitting for training periods with low Y_{sep} . Thus, the absolute error should be a major component based on trials with different combinations of weights.’**

7. ‘How did you set the Adam parameters?’

Reply:

We applied the default parameter values ($\beta_1 = 0.9$, $\beta_2 = 0.999$) of Adam optimizer. Since we found it is capable of carrying out successful prediction, we did not modify these parameters.

8. 'You should compare the proposed method against state-of-the-art competitors.

Otherwise, it is not possible to tell if the approach is "good or bad".'

Reply:

Thank you very much for your vital suggestion. The machine learning study on the prediction of cable separation is scarce. In addition, we realize the prediction for the dynamic behavior of cable separation. Thus, we strengthened this point in the paper.

We added '**Based on the literature review summarised in Table S1 of SI, many efficient processes for cable separation have been developed at a lab scale; however, process modelling was rare, especially for yield prediction by ML.**' was added in Line 79-82 in Page 5.

And, '**Existing studies modelled cable separation processes as polynomial³⁶ or support vector regression³⁷; however, none of these works carry out dynamic modelling of the process, which is innovatively realised by recurrent neural network (RNN) in this study. Previously, we have developed an RNN model combined with the chemical reaction mechanism for predicting the dechlorination degree of PVC in ethylene glycol/NaOH solvent³⁸. However, the separation of PVC and Cu from cable waste is a physical process that requires a more general modelling framework for predicting the dynamic behaviour without chemical kinetics.**' in Line 93-100 in Page 6.

Reviewers' comments:

Reviewer #1 (Remarks to the Author):

Thanks for incorporating all the comments and improving the MS quality and readability. Kindly check if the last section is COnclusions or the Summary. Check the author's guidelines.

Reviewer #2 (Remarks to the Author):

I have read the revised version of the manuscript.
Thank you for including the corrections. As I mentioned earlier, I believe that the article can be published.

Reviewer #3 (Remarks to the Author):

Dear authors, thanks for the detail responses to my comments. Overall, you have address my concerns (e.g., thanks for releasing your implementation, an important step towards reproducibility). However, there is one important aspect that in my opinion is not yet properly addressed. Particularly, the article is entitled "Predicting... on an asynchronous-parallel recurrent neural network", thus I assume that the development of such model plays an important role in the study (i.e., it is one of the contributions of the work). You already provided a clarification of the "asynchronous-parallel" naming, thus the difference between the way you use the terms and the most frequently use of them is clear. However, the real contribution of the proposed approach is not supported by the experimental study carried out. In simple terms, you proposed an improvement of "vanilla RNN", but you do not provide evidence to support that your method really improves RNN (nor any other method). Please, compare your method against state-of-the-art approaches for time series forecasting, or at least show that the "asynchronous-parallel" approach boost the performance of an RNN.

Reviewer#1

1. 'Thanks for incorporating all the comments and improving the MS quality and readability. Kindly check if the last section is CONclusions or the Summary. Check the author's guidelines.'

Reply:

We sincerely appreciate your kind evaluation again. We have checked journal guidelines. And the last section title has been changed into 'Conclusions'.

Reviewer#2

1. 'I have read the revised version of the manuscript.

Thank you for including the corrections. As I mentioned earlier, I believe that the article can be published.'

Reply:

We sincerely appreciate your positive evaluation on the revised paper.

Reviewer#3

1. ‘ Dear authors, thanks for the detail responses to my comments. Overall, you have address my concerns (e.g., thanks for releasing your implementation, an important step towards reproducibility).’

Reply:

Thank you so much for your review on our revised manuscript. We have carefully considered your remaining concerns and try to solve this issue. Please check our response and revision again.

2. 'However, there is one important aspect that in my opinion is not yet properly addressed. Particularly, the article is entitled "Predicting... on an asynchronous-parallel recurrent neural network", thus I assume that the development of such model plays an important role in the study (i.e., it is one of the contributions of the work). You already provided a clarification of the "asynchronous-parallel" naming, thus the difference between the way you use the terms and the most frequently use of them is clear. However, the real contribution of the proposed approach is not supported by the experimental study carried out. In simple terms, you proposed an improvement of "vanilla RNN", but you do not provide evidence to support that your method really improves RNN (nor any other method). Please, compare your method against state-of-the-art approaches for time series forecasting, or at least show that the "asynchronous-parallel" approach boost the performance of an RNN.'

Reply:

You are right that the previous version of paper was lack of the evidence to address the performance of the proposed model. For the simultaneous model training with different series of datasets, some literatures and our previous paper has studied it to obtain the parameters which is applicable to general situations. The universality was achieved by the 'parallel' training, while, the overfitting based on the data points of later period was determined. To solve this issue, the training with asynchronous timesteps is introduced in this paper. We trained the model with fewer number of timesteps (numbers of $dt = 1, 3, \text{ and } 5$ compared with 10 in the main text). We included the validation of those trained models in Fig. S5-7 like Fig. 3 (the original training data and validation data is available at <https://github.com/wilsherelu/Cable-separation->

prediction). The results shows that the fewer timesteps can well converge during the training step, but the errors during the validation are very large. With the increase of timestep numbers, the prediction accuracy can be greatly enhanced.

Thus, we added the text that '**In terms of a classic RNN model for time series prediction, typically only one series of datasets is trained at a time**⁶⁸. **The simultaneous training with multiple series of datasets can facilitate the model to learn general parameters applicable to different series**⁶⁹. **However, there is a phenomenon that the model is overfitted based on data from the later period**³⁷. **Thus, the asynchronous training with multiple timesteps can mitigate the overfitting issue, which can be proven by validating the trained model with fewer timestep numbers shown in Fig. S5-7 compared with Fig. 3.**' in Line 422-428 in page 24-25.

REVIEWERS' COMMENTS:

Reviewer #3 (Remarks to the Author):

Dear authors, I acknowledge the effort made to improve reproducibility of the study. Also, you tried to address the comment on comparing your approach against state-of-the-art approaches. However, to the best of my understanding, the supplementary study includes an ablation study of the time steps, but you are not comparing your method against any other approach. Therefore, we can conclude that it works, and that the selected parameters are "correct", but it is hard to conclude (in my opinion) with the given evidence that the proposed method is improving conventional RNNs. If the focus is the application, that is fine. But if you would like to claim a methodological improvement, you may need to compare the approach to other methods.

Reviewer#3

1. ‘Dear authors, I acknowledge the effort made to improve reproducibility of the study. Also, you tried to address the comment on comparing your approach against state-of-the-art approaches. However, to the best of my understanding, the supplementary study includes an ablation study of the time steps, but you are not comparing your method against any other approach. Therefore, we can conclude that it works, and that the selected parameters are "correct", but it is hard to conclude (in my opinion) with the given evidence that the proposed method is improving conventional RNNs. If the focus is the application, that is fine. But if you would like to claim a methodological improvement, you may need to compare the approach to other methods.’

Reply:

We sincerely appreciate your positive comments on our revised paper. However, based on Fig. S5-7, we not only carried out ‘an ablation study of the time steps’, but also compared our proposed method with the conventional RNN. Because for Fig. S5, there is only one timestep for training and validation, which is not asynchronous. To address this point more clearly, we revised the related sentence in the main text.

We added the text ‘**In addition, the validation of the trained model with one timestep in Fig. S5, which can be regarded as a conventional RNN, elucidates that the prediction cannot be accomplished without asynchronous training.**’ in Line 317-319 in Page 16.